# Constitutive NOS Production Is Modulated by Alzheimer’s Disease Pathology Depending on APOE Genotype

**DOI:** 10.3390/ijms25073725

**Published:** 2024-03-27

**Authors:** Chiara Giuseppina Bonomi, Alessandro Martorana, Denise Fiorelli, Marzia Nuccetelli, Fabio Placidi, Nicola Biagio Mercuri, Caterina Motta

**Affiliations:** 1UOSD Memory Clinic, Policlinico Tor Vergata, University of Rome “Tor Vergata”, 00133 Rome, Italy; chiarag.bonomi@gmail.com (C.G.B.); caterinamotta86@hotmail.it (C.M.); 2Department of Biomedicine and Prevention, University of Rome “Tor Vergata”, 00133 Rome, Italy; fiorellidenise@gmail.com (D.F.); marzianuccetelli@yahoo.com (M.N.); 3Neurology Unit, Policlinico Tor Vergata, University of Rome “Tor Vergata”, 00133 Rome, Italy; fplacidi@gmail.com (F.P.); mercurin@med.uniroma2.it (N.B.M.)

**Keywords:** Alzheimer’s disease, nitric oxide synthase, apolipoprotein E, amyloid β

## Abstract

Both the endothelial (eNOS) and the neuronal (nNOS) isoforms of constitutive Nitric Oxide Synthase have been implicated in vascular dysfunctions in Alzheimer’s disease (AD). We aimed to explore the relationship between amyloid pathology and NO dynamics by comparing the cerebrospinal fluid (CSF) levels of nNOS and eNOS of 8 healthy controls (HC) and 27 patients with a clinical diagnosis of Alzheimer’s disease and isolated CSF amyloid changes, stratified according to APOE ε genotype (APOE ε3 = 13, APOE ε4 = 14). Moreover, we explored the associations between NOS isoforms, CSF AD biomarkers, age, sex, cognitive decline, and blood–brain barrier permeability. In our cohort, both eNOS and nNOS levels were increased in APOE ε3 with respect to HC and APOE ε4. CSF eNOS inversely correlated with CSF Amyloid-β42 selectively in carriers of APOE ε3; CSF nNOS was negatively associated with age and CSF p-tau only in the APOE ε4 subgroup. Increased eNOS could represent compensative vasodilation to face progressive Aβ-induced vasoconstriction in APOE ε3, while nNOS could represent the activation of NO-mediated plasticity strategies in the same group. Our results confirm previous findings that the APOE genotype is linked with different vascular responses to AD pathology.

## 1. Introduction

Alzheimer’s disease (AD) is the most prevalent neurodegenerative disorder among the elderly worldwide. The main neuropathological hallmarks are represented by extracellular senile plaques and intracellular neurofibrillary tangles. The primary pathogenetic hypothesis, rooted in the amyloid cascade theory [1], identified the occurrence of amyloid pathology as the main mechanism driving neuronal dysfunction and leading to cognitive decline [2]. However, other factors could drive neurodegeneration. Indeed, cerebrovascular pathology has often been described as co-occurring pathology in AD brain samples [3]. It has been demonstrated that cerebrovascular changes, whether linked to aging or to the presence of risk factors, e.g., diabetes mellitus, hypertension, atherosclerosis, and atrial fibrillation, potentially and independently contribute to the occurrence of dementia [4]. Thus, the relationship between cerebrovascular dysfunction and AD has gained attention as a key element in the onset and progression of the disease [5], especially at the level of the neurovascular unit (NVU). The NVU constitutes the blood–brain barrier (BBB) and regulates the cerebral blood flow (CBF) to guarantee the supply of oxygen and nutrients and to ensure the clearance of toxic metabolites produced by neurons and glial cells, which are both necessary to support brain health and to meet energy demands. Solid neuropathological evidence demonstrates that both macrostructural and microstructural vascular alterations are linked to diminished CBF and defective BBB permeability, conditions that both weigh on the progression of AD [6,7]. 

An important role in the regulation of blood flow is exerted by nitric oxide (NO), a free radical acting as a gaseous signaling molecule that can passively permeate cellular membranes to generate several pleiotropic effects mainly aimed at immunological regulation and vasodilation [8]. Nevertheless, NO has also been linked with mechanisms of long-term potentiation (LTP) in physiological conditions [9] and excitotoxicity/dysfunction [10] in neurodegenerative disease, suggesting multiple interesting ties with the synaptic failure occurring in AD [11]. NO endogenous production is regulated by three isoforms of synthesizing enzyme, namely the endothelial (eNOS) and neuronal (nNOS) isoforms being constitutively expressed [12]; a third isoform is a Ca^2+^-independent isoform which is generally synthesized following inflammatory stimuli (iNOS) [13].

In recent years, there has been increasing focus on the role of constitutive NOS production as a possible actor in the pathophysiology of AD [14], especially in light of its tight relationship with endothelial and vascular dysfunction [15]. Notably, preclinical studies on animal models show that the endothelial release of NO may affect the function of surrounding brain cells [16,17], and under pathological conditions, the loss of this neurovascular coupling could have detrimental effects, worsening neurodegeneration. Furthermore, while genetic inactivation of eNOS in mice (eNOS^−/−^) has been shown to have direct effects on the shift towards the amyloidogenic pathway of APP processing [18,19], high concentrations of amyloid-β peptides (Aβ) also led to increased formation of reactive oxygen species (ROS). These products, acting directly on pericytes, may, in turn, stimulate their contraction, reducing the caliber of cortical arterioles and capillaries, with detrimental effects on vascular reactivity and CBF. Interestingly, the impact of tau on neurovascular pathology in AD has also been investigated. Pathological tau species, via astrocytic end-feet and interstitial fluids, have the potential to propagate to endothelial cells and pericytes, leading to the disruption of the blood–brain barrier (BBB). Furthermore, it has been demonstrated that tau proteins could support neurovascular pathology in AD through mitochondrial dysregulation and ROS production [20] and, conversely, in murine models of AD, the inactivation of eNOS (APP/preselinin/eNOS^−/−^) increases intraneuronal tau phosphorylation [21].

For these reasons, in the present study, designed to explore the relationship between Aβ pathology and constitutive NOS, we used the recent biomarker-based framework adopted in research and clinics to stratify patients and select our cohort [22]. This framework gave us the opportunity to identify, within the so-called “AD continuum”, patients with isolated amyloid pathology (A+) but still no evidence of significant tau pathology (T−) from those showing full-blown AD (A+T+). Thus, we selected patients with isolated amyloidopathy (A+T−) and directly investigated the possible relationship between constitutive NOS and cerebrospinal fluid (CSF) Aβ levels.

Notably, in our analysis, we considered the role of the apolipoprotein E (APOE) genotype, recognized as the primary genetic risk factor associated with AD, and with multiple crucial implications in the regulation of cerebrovascular-related changes observed in the disease [23]. One of the major functions of apolipoprotein E (ApoE) in the central nervous system is to mediate lipid transport to support cell repair, with the apolipoprotein ApoE4 being less effective than the other isoforms E3 and E2. Of note, the APOE ε4 allele has been associated with a more aggressive course of the disease, enhanced Aβ deposition in the brain, greater cerebral atrophy, and faster cognitive decline [24]. At the same time, APOE ε4 accelerates BBB breakdown in vivo [25,26], facilitates recurrent hemorrhages in cerebral amyloid angiopathy (CAA) [27], and has been associated with reduced CBF [28] and with increased neuropathological evidence of vascular lesions, small vessel disease, and atherosclerosis [29,30,31]. Interestingly, in our previous study, we reported the presence of an inverse relationship between the reduction in cerebrovascular reactivity in response to hypoxia and CSF levels of Aβ42 selectively in carriers of the APOE ε3 genotype, suggesting a deficiency in vascular regulation among APOE ε4 patients [32], strengthening the tight relationship between ApoE4, amyloid peptides, and vascular reactivity.

Thus, in the present study, we aim to directly explore the in vivo expression of NOS isoforms in the CSF of patients with a clinical diagnosis of Mild Cognitive Impairment due to AD [33], accounting for the possible weight of different APOE alleles. Specifically, we selected a cohort of patients with isolated amyloid pathology at CSF analysis [34] without evidence of tauopathy or neurodegeneration.

## 2. Results

### 2.1. CSF Levels of eNOS and nNOS across Groups

The study included a total of 27 patients (13 APOE ε3 and 14 APOE ε4) who were compared with 8 HC. All continuous variables are expressed as means ± standard deviations, as reported in Table 1. The two groups of APOE3 and APOE4 did not differ in terms of age, sex, disease duration, and MMSE (APOE ε3: 22.23 ± 2.71; APOE ε4: 21.71 ± 2.43). The Kruskall–Wallis test was significant for differences in CSF levels of eNOS across groups [H(2) = 10.78, *p* = 0.005], with APOE ε3 patients showing higher values than both HC (p_holm_ = 0.014) and APOE ε4 (p_holm_ = 0.003). CSF levels of nNOS were also significatively different across groups [H(2) = 8.228, *p* = 0.016], being higher in APOE ε3 than in both APOE ε4 (p_holm_ = 0.014) and HC (p_holm_ = 0.014) (see Figure 1).

### 2.2. Correlation Analyses between NOS Species and CSF Aβ42

First, we performed correlation analyses to explore the influence of CSF Aβ42 levels on eNOS and nNOS. Considering the whole sample of AD patients, we retrieved a trend of correlation between CSF Aβ42 and eNOS (rho = −0.342, *p* = 0.087) but no significant correlation with nNOS (rho = −0.134, *p* = 0.515). After stratifying patients for APOE genotype, a strong negative correlation between CSF Aβ42 and eNOS was found in the APOE ε3 subgroup (rho = −0.885, *p* < 0.001) but not in the APOE ε4 (rho = 0.011, *p* = 0.970) (see Figure 2).

As for nNOS, we confirmed the absence of a significant correlation with CSF Aβ42 in either group (APOE ε3: rho = −0.099, *p* = 0.751; APOE ε4: rho = −0.169, *p* = 0.563).

### 2.3. Multivariate Regression Analyses in the APOE Subgroups

We performed multivariate regression analyses to evaluate the associations between AD biomarkers (CSF Aβ42 and p-tau) and CSF levels of eNOS and nNOS, accounting for other possible influencing factors such as age, sex (M = 0, F = 1), Qalb and the degree of cognitive impairment (MMSE). As shown in Table 2, we performed separate regressions considering the whole sample (all patients, *n* = 27) and then stratified patients into APOE ε3 and APOE ε4.

The adjusted analyses highlighted that CSF eNOS levels were inversely associated with CSF Aβ42 in the whole sample of patients (β = −0.592, *p* = 0.008). The association was confirmed with increased effect size and strength of statistical significativity in APOE ε3 carriers (β = −0.698, *p* = 0.003) but was lacking in the APOE ε4 subgroup. Notably, the model also highlighted a trend of positive association with MMSE values in the whole group (β = 0.384, *p* = 0.050) and a trend of negative association with age in APOE ε3 carriers (β = −0.341, *p* = 0.053). No association was found between CSF eNOS and any of the variables in the APOE ε4 group.

On the other hand, considering CSF nNOS, we did not find associations with any covariate in the whole patient group nor in the APOE ε3 subgroup. Instead, in APOE ε4 carriers, CSF nNOS was negatively associated with age (β = −0.684, *p* = 0.007) and with CSF p-tau (β = −0.623, *p* = 0.043), with a trend of positive association with female sex (β = 0.613, *p* = 0.061).

## 3. Discussion

In our cohort, APOE ε3 patients showed increased CSF levels of both eNOS and nNOS with respect to APOE ε4 patients and HC. Interestingly, CSF eNOS was inversely correlated with CSF Aβ42 selectively in APOE ε3. Conversely, CSF nNOS did not correlate with Aβ42 in any of the subgroups but was negatively associated with age and CSF p-tau levels only in the APOE ε4 subgroup. To our knowledge, ours is the first attempt at exploring in vivo CSF levels of eNOS and nNOS in patients with a diagnosis of AD, and our results seem to confirm and expand previous findings on the involvement of NOS in neurodegenerative processes and highlight the importance of APOE genotype, which plays a crucial role in modulating both detrimental and compensatory mechanisms that impact on AD pathophysiology.

First, as mentioned above, both NOS isoforms were found to be increased in the CSF of AD patients within the APOE ε3 subgroup, while their levels were similar in the APOE ε4 subgroup with respect to healthy controls. Considering that decreased production of NO—via reduced activity of constitutive NOS—is a known feature of endothelial dysfunction in physiological aging processes in humans [35], the increase in both eNOS and nNOS in APOE ε3 opens to many possible interpretations and might reflect the activation of a condition-specific compensative or detrimental mechanism activated by endothelial cells in response to protein misfolding in AD. Moreover, in our cohort of APOE ε3, the increase in CSF eNOS was strictly associated with the severity of amyloidopathy and reflected an inverse correlation with decreased CSF Aβ42. A possible reason for this could lie in the effects of pathological levels of Aβ oligomers, which increase alongside the decrease in CSF Aβ42 peptides. Indeed, it has been demonstrated that Aβ oligomers are able to stimulate the release of endothelin-1 from astrocytes and endothelial cells, favoring the activation of contractile pericytes around capillaries, hence, inducing vasoconstriction [36]. Thus, the increase in CSF eNOS observed in our cohort could reflect a compensative vasodilative mechanism put forward to counteract progressive Aβ-induced vasoconstriction in APOE ε3 patients, while such compensation seems to be lacking in carriers of ε4, indicating a possible regulatory role of the APOE genotype on the expression of NOS isoforms in AD. Indeed, previous research supports this hypothesis and shows that circulating ApoE4 inhibits eNOS expression and also has dominant negative effects on the ApoE3-induced stimulation of eNOS activity in case of ε3/ε4 heterozygosis [37,38]. This could mean that APOE ε4 carriers might be constitutively unable to rely on eNOS activation and related effects on vascular tone. Alternatively, Aβ pathology could also cause the downregulation of eNOS through a more intense pro-apoptotic effect on the endothelium in the APOE ε4 subgroup, which could lead to reduced NO production and consequent hampering of endothelium-dependent vasodilation [39]. Overall, these findings indicate a clear and strict relationship between APOE, NO synthesis, and Aβ42, which also appears to be coherent with previous in vivo data from our group showing a direct association between the efficacy of cerebrovascular reactivity, measured with transcranial Doppler, and CSF Aβ42 selectively in APOE ε3 AD patients [32].

Interestingly, previous literature also suggests an aberrant and harmful role of eNOS. Higher NO synthesis has been linked with cytotoxic effects and some experimental models showed that increased eNOS activity might generate superoxide anions that can favor the progression of amyloid pathology [40,41]. Despite the differing and contradictory findings, our results highlight the presence of intrinsically different patterns of NOS dynamic response to Aβ insults, which seem to be driven primarily by the APOE genotype.

At the same time, CSF levels of nNOS are also higher in the APOE ε3 group, while comparable to controls in ε4 carriers, and no relationship with amyloidopathy was retrieved. Instead, an inverse correlation between CSF p-tau levels and age was observed selectively in the APOE ε4 group. nNOS is constitutively expressed in a subgroup of neurons diffusely located in cortical and subcortical nuclei of the brain [42]. nNOS-derived NO acts as a neuronal plasticity mediator [43], with important effects on learning and memory processes [44,45], and interruption of nNOS expression in the hippocampus has been experimentally linked to impaired LTP [46]; thus, a higher expression of nNOS could likely indicate an alteration in NO-related synaptic plasticity mechanisms. Although reasons to explain such an increase are certainly complex, some hypotheses can be formulated. For instance, the increase in nNOS could reflect a more enhanced plasticity in the APOE ε3 subgroup than in APOE ε4, possibly as a response to a supposedly more intense and aggressive pathological degeneration occurring in the first rather than the latter [47]. Alternatively, it may represent a constitutive difference between the two groups in terms of plasticity-related mechanisms so that processes such as, for instance, tau phosphorylation/dephosphorylation would be able to induce more intense compensatory enzyme expression in the APOE ε3 subgroup in order to maintain physiological responses [48]. This view is also supported by previous observations by our group that APOE ε3 carriers with amyloidopathy are more vulnerable to LTP impairment with respect to APOE ε4 [48] and support a strict interplay between nNOS and APOE in the regulation of plasticity mechanisms.

Interestingly, our regression analysis corroborates such a hypothesis. Indeed, we retrieved a negative association of CSF nNOS levels with age and with CSF levels of p-tau, so that CSF nNOS decreases with older age and with increasing tau phosphorylation. Concerning the relationship with age, we speculate that, given the role of nNOS in neurogenesis [49], the inverse relationship could reflect a physiological progressive decrease linked to neuronal senescence in the APOE ε4 subgroup rather than the effect of a pathological cascade of events. Likewise, the detrimental effects mediated by p-tau could cause reduced nNOS production in APOE ε4 and also in patients with still non-pathological CSF p-tau values, suggesting a possible interplay between nNOS dynamics and p-tau in determining early synaptic dysfunction.

We acknowledge that our work has some limitations such as the paucity of the sample size of both patients and HC groups. Moreover, the retrospective design hampers the chance of evaluating the impact of eNOS and nNOS on disease progression and longitudinal cognitive decline. Furthermore, the lack of in vivo measurements of cerebrovascular reactivity limits the chances of reaching definite conclusions on the effects of NOS dynamics on CBF modulation. Despite these limitations, our work offers some interesting perspectives for the potential implementation of our results. For example, it would be interesting to explore NOS dynamics in patients with full-blown AD (A+T+) to verify the effects of tauopathy but also to repeat our analysis in patients with available measurements of cerebrovascular reactivity to test the actual role of NOS in vivo.

## 4. Materials and Methods

### 4.1. Subjects’ Enrollment

Between September 2020 and December 2021, we evaluated a total of 60 outpatients from the UOSD Centro Demenze of the University Hospital “Policlinico Tor Vergata” with a clinical suspicion of AD. After initial diagnostic assessment and neurological examination, we performed a complete neuropsychological examination, including the following cognitive domains: general cognitive efficiency (MMSE), verbal episodic long-term memory (Rey auditory verbal long-term memory: 15-word list immediate and 15-min delayed recall), visuospatial abilities and visuospatial episodic long-term memory (complex Rey’s figure: copy and 10 min delayed recall) and executive functions (phonological word fluency, analogic reasoning, Raven’s colored progressive matrices) [50]. The diagnostic work-up also included brain magnetic resonance imaging, fluorodeoxyglucose positron emission tomography/CT Scan, and a lumbar puncture for diagnostic purposes. Disease duration was calculated using standardized semi-structured questions.

Inclusion criteria for this retrospective observational study were (1) patients satisfying the clinical criteria for MCI due to AD [33] and (2) the presence of amyloidopathy at CSF analysis (A+) [22]. To reduce confounders, we excluded patients with positive biomarkers for tau pathology, i.e., CSF phosphorylated-tau^181^ > 55 pg/mL (T+). Other exclusion criteria were (1) treatment with drugs having possible vasoactive effects (e.g., nitroglycerin, thyroid hormones, phosphodiesterase inhibitors, digoxin), (2) history of acute stroke, (3) Hachinski scale score > 4 or radiological evidence of focal ischemic lesions at MRI, (4) presence of other neurological disorders, (5) hematologic diseases or systemic inflammatory conditions, (6) treatment with antiparkinsonian, antidepressant drugs (e.g., selective serotonin reuptake inhibitors, serotonin and norepinephrine reuptake inhibitors, tricyclic antidepressants, or atypical antidepressants) received within six months before the enrollment. No patient was treated with memantine or acetylcholinesterase inhibitors before the lumbar puncture. Patients’ enrolment procedures are summarized in Figure 3.

Genetic testing for APOE was also performed on all subjects. Our final sample included 27 patients with a clinical diagnosis of AD and isolated amyloid changes without biomarker evidence of tau pathology (A+T−), namely 13 APOE ε3 and 14 APOE ε4.

Furthermore, we also enrolled 8 healthy age-matched controls admitted to the Policlinico Tor Vergata Hospital Emergency Department in the same time window. Upon discharge, all patients had received a diagnosis of either psychogenic symptoms or tensive type headache and infections, cognitive impairment, and other neurological conditions had been ruled out. All subjects showed <4 cells/mmc at CSF cell count and had normal protein values and normal AD biomarkers. Demographics from the patients and control groups are reported in Table 1.

We obtained written consent from all participants and/or legally authorized representatives, and the local ethical committee accounted for the study protocol as an observational retrospective design.

### 4.2. CSF Sampling for AD Biomarkers Analysis and APOE Genotype

All lumbar punctures were performed with a sterile technique between 8 and 10 am, and a sample of 10 mL of CSF was collected for each patient in polypropylene tubes. A total of 2 mL were used for biochemical routine analysis, including cell and protein count. The other 6 mL were centrifuged at 2000× *g* at +4 °C for 10 min. The supernatant was pipetted off, gently stirred, and mixed to avoid potential gradient effects. The CSF was then aliquoted in 1 mL portions and frozen at −80 °C for further analysis. Commercially available kits were used to carry out biochemical analysis (Flex reagent cartridge, Dimension Vista System, Siemens Healthcare Diagnostics GmbH, Munich, Germany). CSF concentrations of AD biomarkers (Aβ42, p-tau, and t-tau) were determined using a sandwich enzyme-linked immunosorbent assay (EUROIMMUN Aβ1-42 levels ELISA©, EUROIMMUN p-Tau (181) ELISA©, EUROIMMUN Total tau ELISA©). Both isoforms of NOS were also determined with commercially available ELISA kits (Human Endothelial Nitric Oxide Synthase and Human Neuronal Nitric Oxide Synthase ELISA Kits©; Cusabio Technology LLC, Houston, TX, USA), as per the manufacturer’s instructions. Blood samples were also drawn for complimentary analysis—i.e., CSF/serum albumin quotient (QAlb) accounting for BBB—and APOE genotyping, which was conducted by allelic discrimination technology (TaqMan; Applied Biosystems; Waltham, MA, USA).

### 4.3. NOS Analysis

The ELISA kits “Human Endothelial Nitric Oxide Synthase eNOS” (Cusabio Technology LLC, Houston, TX, USA) and “Human Neuronal Nitric Oxide Synthase nNOS” (Cusabio Technology LLC, Houston, TX, USA) were used for the quantitative determination of human endothelial nitric oxide synthase (eNOS) and human neuronal nitric oxide synthase (nNOS) concentrations in the CSF. These ELISA tests employ the quantitative sandwich immunoenzymatic technique. CSF samples were processed according to the manufacturer’s instructions, and a dilution (1:20) was performed for nNOS before testing. Subsequently, the standard vial was centrifuged at 6000–10,000 rpm for 30 s, and the standard was reconstituted with 1.0 mL of sample diluent. This reconstitution produced a stock solution of 10 IU/mL for nNOS and 70 IU/mL for eNOS. A 250 μL of sample diluent was pipetted into each tube (S0–S6) using the stock solution to produce a series of twofold dilutions. The undiluted standard served as the high standard (10 IU/mL for nNOS and 70 IU/mL for eNOS). The sample diluent served as the zero standard (0 IU/mL). Subsequently, 100 μL of standard and sample were added to each well and incubated for 2 h at 37 °C. The liquid from each well was removed without washing, and then 100 μL of biotinylated antibody (1×) was added to each well and incubated for 1 h at 37 °C. Each well was aspirated and washed, repeating the process twice for a total of three washes. Then, 100 μL of HRP-avidin (1×) was added to each well and incubated for 1 h at 37 °C. The aspiration/washing process was repeated five times. Then, 90 μL of TMB substrate was added to each well and incubated for 15–30 min at 37 °C, protecting the plate from light. Finally, 50 μL of stop solution was added to each well, gently tapping the plate to ensure mixing. The optical density of each well was then determined within 5 min using a microplate reader set at 450 nm. To calculate the results, as per the manufacturer’s instructions, the mean of duplicate readings for each standard and sample was calculated, and the average OD of the zero standard was subtracted. Finally, a standard curve was created by reducing the data with GraphPad Prism Software 10.1.2 (San Diego, CA, USA), capable of generating a four-parameter logistic curve (4-PL). The data were expressed as international units (IU/mL).

### 4.4. Data Management and Statistical Analysis

All continuous variables, including levels of CSF biomarkers (i.e., Aβ42, p-tau, t-tau, eNOS, and nNOS), disease duration, MMSE, age, and Qalb, were expressed in terms of mean ± standard deviations (see Table 1). Patients were stratified in the APOE ε4 subgroup when carrying either one (APOE ε3/ε4) or two (APOE ε4/ε4) alleles. The remaining patients were all APOE ε3/ε3 (APOE ε3 subgroup).

Since data were not normally distributed per the Shapiro–Wilk test, the comparison between groups was performed via the Kruskal–Wallis test for independent samples and using Dunn’s post-hoc test for multiple comparisons, while categorical variables were compared with Pearson’s chi-squared test. We used Spearman’s rho analysis to test the correlation between CSF Aβ42, eNOS, and nNOS. Then, we used multivariate regression analysis to verify the effects of covariates (age, sex, Qalb, MMSE, Aβ42, and p-tau) on CSF levels of eNOS and nNOS, computed as individual dependent variables, in the whole patient sample. The analysis was also repeated after stratifying patients in two independent subgroups, APOE ε3 and APOE ε4.

Statistical analysis and data management were operated via JASP© (Version 0.14—Computer Software—JASP TEAM 2020) and GraphPad Prism© version 10.1.2 for Windows (GraphPad Software, San Diego, CA, USA, www.graphpad.com; accessed on 19 February 2024). All results were computed with two-tailed significativity tests; the significance level for all analyses was set at α = 5%, corresponding to a threshold *p* of <0.05.

## 5. Conclusions

In conclusion, our results confirm and expand previous findings on the importance of APOE-dependent NOS synthesis regulation in AD. Indeed, the APOE genotype serves as a pivotal factor in regulating both adverse and adaptive processes likely influencing vascular remodeling in AD. If confirmed, these results might pave the way for the use of NOS isoforms as dynamic fluid biomarkers of cerebral blood flow regulation and for their potential integration into the diagnostic framework for AD. Evaluating the cerebrovascular compartment at the single-patient level could help the advancement of personalized therapeutic strategies aimed at addressing the unique needs of each patient. This approach holds promise for enhancing treatment efficacy and optimizing patient outcomes.

## Figures and Tables

**Figure 1 ijms-25-03725-f001:**
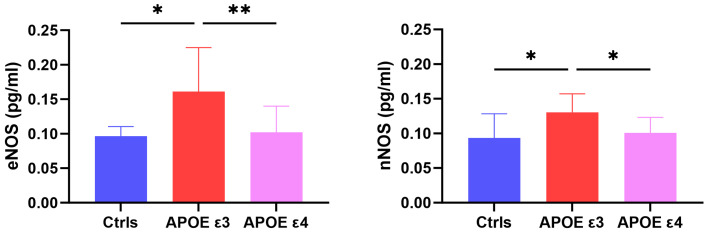
Results from the comparison of CSF levels of eNOS (**left**) and nNOS (**right**) across groups (* ≤ 0.05; ** ≤ 0.01).

**Figure 2 ijms-25-03725-f002:**
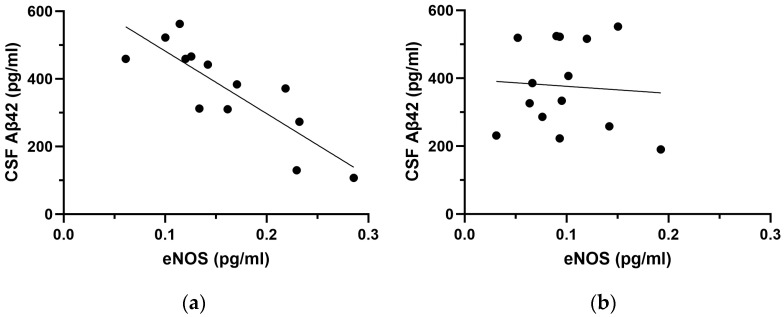
Scatter plots showing the correlation between CSF eNOS levels and Aβ42 in the APOE ε3 (**a**) and in the APOE ε4 subgroup (**b**).

**Figure 3 ijms-25-03725-f003:**
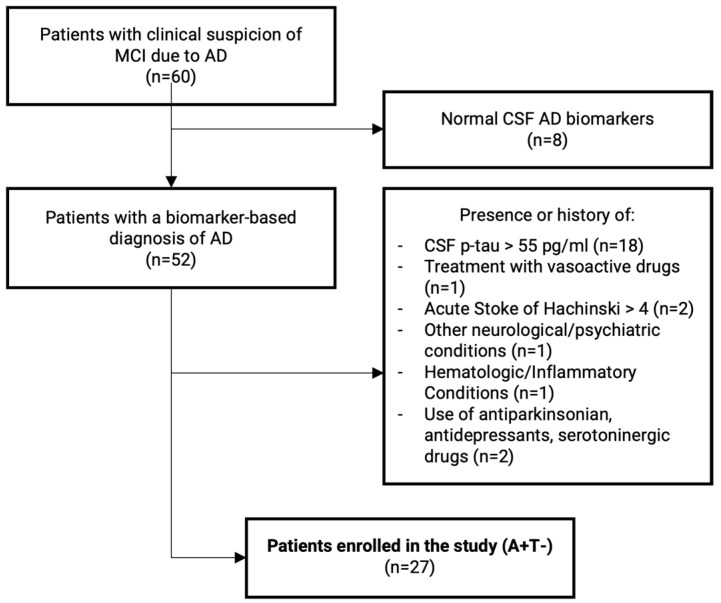
Flowchart summarizing patient selection procedures for the study.

**Table 1 ijms-25-03725-t001:** Demographics and results of CSF analyses in groups, expressed as means ± standard deviations.

	HC (*n* = 8)	APOE ε3 (*n* = 13)	APOE ε4 (*n* = 14)	*p*
Age	67.25 ± 10.36	72.62 ± 7.33	70.50 ± 8.14	0.400
Sex (%F)	62.5%	46.15%	35.71%	0.479
Disease Duration	n.a.	16.2 ± 4.2	15.9 ± 5.1	0.672
MMSE	28.62 ± 1.06	22.23 ± 2.71	21.71 ± 2.43	**<0.001 *****
Qalb	6.97 ± 2.72	6.35 ± 2.86	7.35 ± 2.52	0.619
Aβ42 (pg/mL)	919.21 ± 329.22	369.16 ± 139.63	376.74 ± 130.36	**<0.001 *****
Aβ40 (pg/mL)	5417.03 ± 1808.92	3917.48 ± 2515.43	4519.43 ± 1845.67	0.159
p-tau (pg/mL)	29.81 ± 13.46	34.00 ± 18.39	35.32 ± 16.97	0.767
t-tau (pg/mL)	205.14 ± 112.18	210.17 ± 103.82	207.06 ± 101.77	0.901
eNOS (pg/mL)	0.097 ± 0.014	0.161 ± 0.064	0.098 ± 0.042	**0.005 ****
nNOS (pg/mL)	0.094 ± 0.036	0.131 ± 0.026	0.100 ± 0.023	**0.016 ***

HC: healthy controls; F: female; MMSE: mini-mental state examination; Qalb: albumin quotient; NOS: nitic oxide synthase; eNOS: endothelial NOS; nNOS: neuronal NOS. Bold values represent statistical significativity (* ≤ 0.05; ** = <0.01; *** ≤ 0.001); n.a.: not available.

**Table 2 ijms-25-03725-t002:** Results from the stepwise backward multivariate regression analyses in patients.

		All Patients (*n* = 27)		APOE ε3 (*n* = 13)		APOE ε4 (*n* = 14)	
eNOS (pg/mL)		β	*p*	β	*p*	β	*p*
	**Age**	−0.195	0.291	−0.341	0.053	−0.511	0.138
	**Sex**	−0.162	0.477	0.070	0.797	−0.048	0.920
	**Qalb**	−0.241	0.293	−0.001	0.997	−0.182	0.608
	**MMSE**	0.384	0.050	0.251	0.295	−0.050	0.914
	**Aβ42**	−0.592	**0.008 ****	−0.698	**0.003 ****	−0.165	0.750
	**p−tau**	−0.276	0.163	0.145	0.433	−0.552	0.237
R^2^		0.440		0.901		0.509	
**nNOS (pg/mL)**		**β**	** *p* **	**β**	** *p* **	**β**	** *p* **
	**Age**	−0.258	0.235	−0.227	0.576	−0.684	**0.007 ****
	**Sex**	0.105	0.691	−0.262	0.723	0.613	0.061
	**Qalb**	−0.377	0.163	−0.805	0.386	−0.088	0.668
	**MMSE**	−0.097	0.656	0.164	0.790	−0.038	0.887
	**Aβ42**	0.006	0.978	0.120	0.766	−0.170	0.575
	**p-tau**	−0.120	0.597	−0.007	0.988	−0.623	**0.043 ***
R^2^		0.234		0.276		0.835	

Qalb: albumin quotient; MMSE: mini-mental state examination; NOS: nitic oxide synthase; eNOS: endothelial NOS; nNOS: neuronal NOS. Bold values represent statistical significativity (* ≤ 0.05; ** ≤ 0.01).

## Data Availability

Data are available upon reasonable request to the authors.

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
