# Peer review of "Constitutive NOS Production Is Modulated by Alzheimer’s Disease Pathology Depending on APOE Genotype"

_ijms, 2024, doi:10.3390/ijms25073725_

Round 1
Reviewer 1 Report
Comments and Suggestions for Authors
Dear Authors,
I have carefully read your paper, which goal was to explore the in-vivo expression of NOS isoforms (eNOS and nNOS) directly in CSF samples of a cohort of patients with a clinical diagnosis of Mild Cognitive Impairment due to AD and isolated amyloid pathology at CSF analysis, thus without evidence of tauopathy or neurodegeneration. APOE genotyping was also performed to evaluate the effects of different alleles (ε3 and ε4) on this relationship.
The topic has great scientific relevance and there are no similar papers published so far which adds value to the paper.
However paper has flaws which need to be corrected in order to improve the paper quality:
- Add aim in abstract
- In the last paragraph of Introduction, you wrote the aim of the paper. Please try to make it more concise.
- The 2.section should be Methodology, Results are the third and Discussion is the forth – change this
- In the methods, state what type of study this is. Is this case-control? Is it retrospective or prospective?
- Add Flow chart in the Methods to clearly state how you obtained study sample, from 60 to 27 patients
- State inclusion and exclusion criteria
- Explain how did you match controls? Why there was not the same number of controls as cases?
- In the discussion, explain your findings in the first paragraph
- Add limitations in the last paragraph of the discussion
- Add conclusion as the 5th section of the paper.
I suggest to revise the paper.
Comments on the Quality of English LanguageMinor changes needed
Author Response
Response to Reviewer #1
- Add aim in Abstract
- In the last paragraph of Introduction, you wrote the aim of the paper. Please try to make it more concise
We thank the reviewer for these observations, we added the main aim in the abstract (see page 1, lines14-15) and shortened the last section of the introduction: “Thus, in the present study we aimed at directly exploring the in-vivo expression of NOS isoforms in the CSF of patients with a clinical diagnosis of Mild Cognitive Impairment due to AD [31], accounting for the possible weight of different APOE alleles. Specifically, we selected a cohort of patients with isolated amyloid pathology at CSF analysis [32] without evidence of tauopathy or neurodegeneration” (see page 2, lines 102-106)
- The second.section should be Methodology, Results are the third and Discussion is the fourth – change this
The manuscript was formatted according to the journal’s instruction, which require to place the Methodology Section after the Results and Discussion section. If this is fine with the Editor, we would be happy to change the order of the sections.
- In the methods, state what type of study this is. Is this case-control? Is it retrospective or prospective?
- Add Flow chart in the Methods to clearly state how you obtained study sample, from 60 to 27 patients
- State inclusion and exclusion criteria
Thank you for pointing these elements out, we carefully reviewed the Methods section to add the required details on the study design as well as to clarify inclusion/exclusion criteria: “Inclusion criteria for this retrospective observational study were: (1) patients satisfying the clinical criteria for MCI due to AD [49] and (2) the presence of amyloidopathy at CSF analysis (A+) [20]. To reduce confounders, we excluded patients with positive biomarkers for tau pathology, i.e. CSF phosphorylated-tau181 > 55 pg/ml (T+). Other exclusion criteria were: (1) treatment with drugs having possible vasoactive effects (e.g. nitroglicerine, thyroid hormones, phosphodiesterase inhibitors, digoxin), (2) history of acute stroke, (3) Hachinski scale score >4 or radiological evidence of focal ischemic lesions at MRI, (4) presence of other neurological disorders, (5) hematologic diseases or systemic inflammatory conditions, (6) treatment with antiparkinsonian, antidepressant drugs (e.g., Selective Serotonin Reuptake Inhibitors, Serotonin and Norepinephrine Reuptake Inhibitors, Tricyclic antidepressants or Atypical antidepressant) received within six months before the enrollment. No patient was treated with memantine or acetylcholinesterase inhibitors before the lumbar puncture” (lines 266-279, page 7). Moreover, as requested, we included a Flowchart to summarize the enrolment procedures that were used for the study (see Figure 3), explaining how the sample was reduced from 60 to 27.
- Explain how did you match controls? Why there was not the same number of controls as cases?
Our study compared the CSF concentrations of constitutive NOS isoforms of patients with AD and healthy controls (HC). Since the CSF is obtained through a lumbar puncture (LP), which is an invasive procedure, the availability of these samples in HC is limited. The study enrolled subjects undergoing LP for other medical reasons as per normal clinical practice, and for which a diagnosis of neurological conditions had been already ruled out. Hence, we were not able to enroll as many HC as patients with AD. The Kruskal-Wallis test accounts for sample numerosity, thus we are confident with the results obtained. We mentioned this issue in the limitations section.
- In the discussion, explain your findings in the first paragraph
We added a short summary of our findings at the beginning of the Discussion section: “In our cohort APOE ε3 patients showed increased CSF levels of both eNOS and nNOS with respect to APOE ε4 patients and HC. Interestingly, CSF eNOS was inversely correlated with CSF Aβ42 selectively in APOE ε3. Conversely, CSF nNOS did not correlate with Aβ42 in any of the subgroups but was negatively associated with age and CSF p-tau levels only in the APOE ε4 subgroup” (see page 5, lines 159-163).
- Add limitations in the last paragraph of the discussion
Following the reviewer’s suggestion we clarified the limitations of our study in the last paragraph of the discussion: “We acknowledge that our work has some limitations such as the paucity of the sample size of both patients and HC groups. Moreover, the retrospective design hampers the chance of evaluating the impact of eNOS and nNOS on disease progression and longitudinal cognitive decline. Furthermore, the lack of in-vivo measurements of cerebrovascular reactivity limits the chances of reaching definite conclusions on the effects of NOS dynamics on CBF modulation” (see page 6, lines 238-243).
- Add conclusion as the 5thsection n of the paper.
Following the reviewer’ instructions, we added a fifth “Conclusions” section to the manuscript: “In conclusion, our results confirm and expand previous findings on the importance of the APOE-dependent NOS synthesis regulation in AD. Indeed, APOE genotype serves as a pivotal factor in regulating both adverse and adaptive processes likely influencing vascular remodeling in AD. If confirmed, these results might pave the way for the use of NOS isoforms as dynamic fluid biomarkers of cerebral blood flow regulation and for their potential integration into the diagnostic framework for AD. Evaluating the cerebrovascular compartment at the single patient level could help toward the advancement of personalized therapeutic strategies aimed at addressing the unique needs of each patient. This approach holds promise for enhancing treatment efficacy and optimizing patient outcomes” (see page 9, lines 365-374).
Reviewer 2 Report
Comments and Suggestions for Authors
Dear authors, I enjoyed reading the manuscript entitled "Constitutive NOS production is modulated by Alzheimer's Disease pathology depending on APOE genotype". The original paper, based on a clinical trial, explored the levels of eNOS and nNOS in CSF in patients with a diagnosis of AD.
The manuscript complies with the requirements of the journal, the bibliographic references are appropriate for the presented topic. The results obtained and reproduced through the 2 tables and figures, offer the possibility of making the reading of the manuscript easier for the readers.
However, I have a number of questions and suggestions:
1. We noted that only female patients were included in the study. Is there any motivation why this gender was preferred?
2. The age of the patients in the study is generally between 60-70 years. Alzheimer's dementia occurs by this age. Would it be possible for the authors to conduct a study with younger patients?
3. The number of bibliographic references is small. I ask the authors to consult the specialized literature and add specific references with the presented subject.
4. The study is from the clinical area and I really appreciate that. Alzheimer's dementia is currently being studied quite a bit, trying to find possible causes. As a rule, clinical studies start on the basis of preclinical studies. In terms of preclinical studies, do the authors have knowledge of this topic? Could you make a small comparison between preclinical and clinical, related to eNOS and nNOS expression?
5. In the discussion chapter, the conclusions are also presented. I ask the authors to insert a Conclusions only chapter for clarity.
6. The English language could be improved.
Comments on the Quality of English Language
The English language could be improved.
Author Response
Response to Reviewer #2
- We noted that only female patients were included in the study. Is there any motivation why this gender was preferred?
The present study included both female and male subjects. Specifically, the HC group (n=8) included 5 females (62.5%) and 3 males; the APOE e3 subgroup (n=13) included 6 females (46.15%) and 7 males; the APOE e4 subgroup (n=14) included 5 females (35.71%) and 9 males. As reported in Table 1, no significant difference was found in terms of frequency between the three cohorts (p=0.479).
- The age of the patients in the study is generally between 60-70 years. Alzheimer's dementia occurs by this age. Would it be possible for the authors to conduct a study with younger patients?
We thank the reviewer for this suggestion. Our study had a retrospective design and enrolled 60 consecutive patients with a clinical suspicion of MCI due to AD, which as pointed out by the reviewer usually debuts in subjects aged between 60 and 70. Unfortunately, we have no data available on younger patients at the moment, but we will try to seize this suggestion in future research designs.
- The number of bibliographic references is small. I ask the authors to consult the specialized literature and add specific references with the presented subject.
- The study is from the clinical area and I really appreciate that. Alzheimer's dementia is currently being studied quite a bit, trying to find possible causes. As a rule, clinical studies start on the basis of preclinical studies. In terms of preclinical studies, do the authors have knowledge of this topic? Could you make a small comparison between preclinical and clinical, related to eNOS and nNOS expression?
We firmly believe in the role of preclinical studies to support clinical evidence. Following the reviewer’s suggestion, in the text we expanded the text to specify which among the included references was referring to preclinical studies: “Notably, preclinical studies on animal models show that the endothelial release of NO may affect the function of surrounding brain cells [16,17] and, under pathological con-ditions, the loss of this neurovascular coupling could have detrimental effects worsening neurodegeneration. Furthermore, while genetic inactivation of eNOS in mice (eNOS-/-) has been shown to have direct effects on the shift towards the amyloidogenic pathway of APP processing [18,19], high concentrations of amyloid-β peptides (Aβ) also led to increased formation of reactive oxygen species (ROS) […] and, conversely, in murine models of AD the inactivation of eNOS (APP/preselinin/eNOS-/-) increases intraneuronal tau phosphorylation [21]” (see page 2 lines 61-68 and 75-77). We also added a citation (see ref. #21).
- In the discussion chapter, the conclusions are also presented. I ask the authors to insert a Conclusions only chapter for clarity.
Following the reviewer’s instructions, we added a fifth “Conclusions” section to the manuscript: In conclusion, our results confirm and expand previous findings on the importance of the APOE-dependent NOS synthesis regulation in AD. Indeed, APOE genotype serves as a pivotal factor in regulating both adverse and adaptive processes likely influencing vascular remodeling in AD. If confirmed, these results might pave the way for the use of NOS isoforms as dynamic fluid biomarkers of cerebral blood flow regulation and for their potential integration into the diagnostic framework for AD. Evaluating the cerebrovas-cular compartment at the single patient level could help toward the advancement of personalized therapeutic strategies aimed at addressing the unique needs of each patient. This approach holds promise for enhancing treatment efficacy and optimizing patient outcomes.” (see page 9, lines 365-374).
- The English language could be improved.
As requested, the English language was carefully revised throughout the text.
Round 2
Reviewer 1 Report
Comments and Suggestions for Authors
Dear authors,
Thank you for revising the paper according to my suggestions. The quality of the paper is significantly approved.